# Spicy Bitumen: Curcumin Effects on the Rheological and Adhesion Properties of Asphalt

**DOI:** 10.3390/ma14071622

**Published:** 2021-03-26

**Authors:** Abraham A. Abe, Cesare Oliviero Rossi, Paolino Caputo, Maria Penelope De Santo, Nicolas Godbert, Iolinda Aiello

**Affiliations:** 1Dipartimento di Chimica e Tecnologie Chimiche, Università della Calabria, 87036 Arcavacata di Rende, Italy; abraham.abe@unical.it; 2Dipartimento di Fisica e CNR-Nanotec, Università della Calabria, via Bucci 31C, 87036 Rende, Italy; maria.desanto@fis.unical.it; 3MAT-InLAB, LASCAMM CR-INSTM, Unità INSTM della Calabria, Dipartimento di Chimica e Tecnologie Chimiche, Università della Calabria, Ponte Pietro Bucci Cubo 14C, 87036 Rende, Italy; nicolas.godbert@unical.it (N.G.); iolinda.aiello@unical.it (I.A.); 4CNR NANOTEC-Istituto di Nanotecnologia U.O.S. Cosenza, 87036 Rende, Italy

**Keywords:** road pavements, rheological modifiers, adhesion promoters, anti-oxidant agents, bitumen, turmeric, curcumin

## Abstract

Over the years, the need for the synthesis of biodegradable materials has facilitated the drift of the asphalt industry towards eco-sustainable and cost-effective production of road pavements. The principal additives in the asphalt industry to improve the performance of road pavements and increase its lifespan are majorly rheological modifiers, adhesion promoters and anti-oxidant agents. Rheological modifiers increase physico-chemical properties such as transition temperature of asphalt binder (bitumen), adhesion promoters increase the affinity between binder and stone aggregates while anti-oxidant agents reduce the effects of oxidation caused by exposure to air, water and other natural elements during the production of asphalt pavements. In this study, we tested the effectiveness of a food grade bio-additive on these three aforementioned properties. We also sought to hypothesize the mechanisms by which the additive confers these desired features on bitumen. We present this study to evaluate the effects of turmeric, a food-based additive, on bitumen. The study was conducted through dynamic shear rheology (DSR), atomic force microscopy, scanning electron microscopy (SEM) and boiling test analysis.

## 1. Introduction

Bitumen is a black, soft, viscoelastic material, consisting of several components (predominantly hydrocarbons and their derivatives). It is obtained by fractional distillation during petroleum industry refinery processes of crude oil. Bitumen is a colloidal dispersion of asphaltenes in a continuous (maltenic) phase composed of aromatic compounds, saturated paraffins, and resins. However, the specific chemical composition of bitumen is highly related to both the nature of the initial crude oil used as starting material and the cracking process carried out on it. Bitumen is employed globally in asphalt mixes as the main binder for road pavement construction. Due to the viscoelastic properties of bitumen, its mechanical response is both temperature and time dependent. However, due to the hugely significant variation in climatic conditions on a geographical basis, asphalt materials need to be selected carefully in order to increase the longevity and lifespan of the road pavement thus reducing the possible huge cost of road maintenance [1]. The durability and sustainability of bitumen is a concept based on the resistance to variations of its physical properties due to environmental exposure. At room temperature, bitumen is chemically stable and inert and when exposed to the atmosphere as a thin film, bitumen starts hardening quickly becoming very fragile. Therefore, different modifiers and additives have been incorporated into bitumen in order to enhance its performance [2,3,4,5,6,7]. However, not all of these additives can be considered as “green” additives that would in addition to their properties safeguard the environment.

The main physical change of bitumen occurs as a rheological variation, with a systematic increase in stiffness. This phenomenon is partly due to the loss of bitumen’s lighter fractions during manufacturing or mixing processes with stone aggregates.

Common additives such as polyphosphoric acid react chemically with bitumen to improve its rheological properties [8,9]. Polymers such as poly(styrene-butadiene-styrene), (SBS) also improve rheological properties but they do so physically by imparting their own properties into the matrix thus improving its properties [10,11]. Adhesion promoters such as amides and hydrated lime either chemically interact with bitumen to reduce the surface free energy of the constituents of the asphalt conglomerate or increase water resistance of asphalt by preventing water molecules from getting into the bituminous matrix [12,13]. Rejuvenating agents are used to recycle the old road pavements (reclaimed asphalt pavement otherwise known as RAP) due to their ability to interact with aged bitumen aggregates to regenerate the original unaged bitumen structure [14,15,16,17].

Anti-oxidant agents function by taking part in the oxidation process themselves thereby preventing or reducing the possibility of oxidation on the fundamental constituents of the asphalt pavement itself. These agents reduce changes in bitumen’s chemical structure which are caused by its reaction with atmospheric oxygen [7,18].

The aforementioned additives, even though are proven to be effective, generally are not eco-friendly as they are either too acidic, too basic or in one way or another, adversely affect the environment in which the road is paved. As a matter of fact, recently some bio-polymers are starting to get tested and utilized as modifiers in road paving [19,20].

What this study aims at is to achieve the improvement of rheological properties, adhesion promotion and anti-oxidant properties of road pavements in an eco-friendly way.

Turmeric, obtained from the rhizomes of *Curcuma longa* L., is one of the main culinary spices in Asian cooking—especially Indian. It is also known as the golden spice, denoted by different names in different languages worldwide and is an important ingredient in curry. It is known for its bright yellow colour and is also added in European cuisine to enhance the colour of the prepared dish [21,22]. The orange-yellow colour and taste of turmeric rhizome is attributed to the presence of curcuminoids—which are regarded and accepted as one of its major biomolecular components (3–5%) [23]. Chemically, in relation to this type of turmeric constituents, all curcuminoids feature a conjugated system with a characteristic skeleton of two aromatic rings joined by an aliphatic chain—usually a heptane chain and in some cases, a 1,3-diketone group. Generally, the principal constituent of these three polyphenols, is curcumin (1) which is widely used as food colouring, preservative, additive and flavour agent (E100). Demethoxycurcumin (2) and bis-demethoxycurcumin (3) in which one or both -OCH3 functionalities over phenol rings are removed, are the minor curcuminoid components (Figure 1) [23,24]. Turmeric, the “natural and commercial curcumin” is the powder obtained from the dried and crushed rhizomes of Curcuma longa L. For this reason, the main part of the product is indeed of a carbohydrate nature (ca. 70%) [25].

Only ca. 20 to 25% of turmeric powder can contain curcuminoid derivatives which typically are a mixture of three curcuminoids, whose composition of is about 70% curcumin (curcumin I) (1), 19% demethoxycurcumin (2), and 9% bisdemethoxycurcumin (3) [26,27]. Curcumin was first isolated in 1815, but only in 1910 was the exact chemical structure depicted [28]. Pabon however, successfully prepared this molecule for the first time in excellent yield in 1964 [29]. In recent years, curcuminoids and in particular, curcumin have attracted a lot of attention due to their pharmacological activity and biological effects, tested directly or chelated to metal centres [30,31,32,33]. Furthermore, curcumin has been deeply studied for its antioxidant properties in molecular biology research [34,35]. Previous work carried out by our group already demonstrated that carbohydrate rich compounds such as flours, extracts from vegetable products and gums containing polysaccharides demonstrated that polysaccharides increase the rigidity of bitumen and improve the mechanical properties of bitumen with an increase in temperature [36]. For these reasons, it was expected that turmeric could be a good candidate to be tested in bitumen matrix as a multi-functional agent. Indeed, turmeric has the right chemico-physical characteristics to be a multifunctional bio-additive that will act as a chemical modifier improving the anti-oxidant properties of the bitumen binder, improving the asphalt concrete mechanical properties and finally that might also promote adhesion between the bitumen binder and the aggregates. Owing to the multi-component nature of turmeric (food grade curcuma, FGC), two different pure grades of curcumin were also investigated in order to examine the effects of curcuminoids on the bitumen properties. Thus, the functionality and effectiveness of each one was determined. The reagent grade curcumin (RGC) which contains ca. 20% of various methoxy-derivatives of curcumin was used since its composition is close to the natural composition in curcuminoids present in turmeric. A high purity grade curcumin (HPGC) was also used as reference to probe the effect of the main active compound (Figure 2).

Furthermore, since bitumen is also a complex matrix, two different kinds of bitumen were probed. The bitumen samples were modified at various percentages to effectively evaluate the extent of the investigated mechanism. To determine the sol-transition temperature of the modified bitumen samples, time-cure rheological tests were carried out [8]. An empirical standard method—the boiling test (Riedel and Wieber test) [13] was also performed in order to test the adhesion capacity of the modified bitumen. To evaluate the changes in the morphology of the bitumen with the addition of the new additives, Atomic Force Microscopy (AFM) and Scanning Electron Microscopy (SEM) sample analysis were carried out [37,38].

## 2. Experimental Section

### 2.1. Materials

The analyzed bitumens LC50 and C170 were supplied by Loprete Costruzioni Stradali (Terranova Sappo Minulio, Italy). Both were used as received. The crude oil was from Saudi Arabia. Their penetration grades are 50/70, for LC50, and 170/210, for C170. The penetration grade was determined by the ASTM D946 standardized protocol [39] which entails a standard needle being loaded with a 100 g weight and the length covered by the needle in the bitumen sample is measured in tenths of a millimeter for a known time, at fixed temperature. An acidic stone (commercial name: “Porfido del Trentino”) was chosen as the tested stone material for boiling tests and was supplied by “Porfido Trentino SRL” company (Albiano, Italy) [7]. CaCO_3_ was also tested as an inert filler for the purpose of comparison.

Curcumin (HPGC) was obtained from Biosynth Carbosynth (Enching, Germany) (purity HPLC min 95%); curcumin (RGC) was obtained from Merck Sigma Aldrich (Milano, Italy) (purity for HPLC ≥ 75.0% (a/a), Bisdemethoxycurcumin ≤ 5.0% (a/a), Demethoxycurcumin ≤ 20.0% (a/a)). Turmeric (FGC) was obtained from the local supermarket.

The commercial bitumens used in this work mainly differ by their penetration value—the first one with a low value of 68 dmm, the second one with a higher value of 185 dmm (Table 1). They were selected to determine if their physico-chemical properties might be differently affected by the tested additives, owing to the difference in their chemical composition.

### 2.2. Sample Preparation

All three additives (HPGC, RGC and FGC) are orange solid powders. They were individually mixed with hot bitumen (about 150 °C) within 1–3 wt% ratio. The bitumen modification was carried out by using a mechanical stirrer. Initially, 100 g of bitumen was heated up to 150 °C until it totally melted and flowed freely (about 30 min). Then 1g of additive was added to the melted sample. Meanwhile, the blend was being mixed by a high-speed shear mixer at between 500–700 rpm. The compound was kept under the mixer (and obviously kept on the heater as well to maintain the compound at 150 °C) for 1 h. After the mixing process, some drops of the resulting modified bitumen were laid on a greaseproof paper for further analysis.

### 2.3. Rheological Analysis

Dynamic Shear Rheology (DSR) analysis was performed using a Dynamic Stress Rheometer (SR-5000, Rheometrics Scientific, Piscataway, NJ, USA) equipped with a parallel plate geometry with 25 mm of diameter, using a 2 mm gap. The measurements were carried out in a temperature range of 25–130 °C (increasing at 1 °C/min). The frequency was set at 1 Hz and the Stress at 100 Pa. All experiments were performed in the viscoelastic regime. More details are described in the reference [37].

### 2.4. Boiling Test

The boiling test procedure used in this study was performed according to ASTMD3625. All details can be found in references [7,13].

### 2.5. Atomic Force Microscopy (AFM)

Atomic force microscopy (AFM) studies were performed using a Nanoscope VIII Bruker microscope which was set to operate in tapping mode with oscillations of the cantilever regulated close to its resonance frequency (150 kHz). Due to the fact that the cantilever oscillates up and down, the tip touches the surface of the sample sporadically. As the tip descends near the surface, the interaction between the tip and the surface influences the vibration of the cantilever. The phase angle shift of the cantilever vibration signifies dissipation of energy in the tip-sample ensemble, so it depends on the specific mechanical properties of the sample under testing. Cantilevers with elastic constants of 5 N/m and 42 N/m were used for the measurements. Antimony doped silicon probes (TAP150A, TESPA-V2, Bruker) with resonance frequencies 150 kHz and 320 kHz respectively and nominal tip radius with a 10nm curvature were used. Phase and topography images were taken contemporarily.

### 2.6. Thermogravimetric Analysis (TGA)

Thermogravimetric studies were performed on a Perkin-Elmer TGA-6 instrument. Analysis was performed on ca. 3 to 5 mg of sample. Thermal decomposition was studied from 30 °C to 850 °C under constant nitrogen flux, at a heating rate of 10 °C min^−1^.

### 2.7. Scanning Electron Microscopy (SEM)

Scanning Electron Microscopy (SEM) images were acquired on a Phenom ProX microscope (Thermo Fisher Scientific Inc., Waltham, MA, USA), equipped with a backscattered electron detector for energy dispersive X-ray spectroscopy (EDX). Samples were placed on carbon-conductive, double-coated tabs.

### 2.8. Nuclear Magnetic Resonance Spectroscopy (NMR)

^1^H-NMR spectra were recorded on a Bruker WM-300 (Billerica, MA, USA) (CDCl_3_ solution, internal standard Me_4_Si). For the turmeric powder, an extraction in chloroform was performed prior to analysis, after filtration and evaporation of the solvent the residue was dissolved in deuterated chloroform and the spectrum was registered.

## 3. Results and Discussion

The three additives (HPGC, RGC and FGC) were briefly analyzed. For HPGC and RGC samples, ^1^H-NMR spectra were recorded by dissolving 5 mg of each compound in 1 mL of deuterated chloroform. For FGC, ^1^H-NMR was performed after extraction. Two grams of FGC were extracted with 500 mL of chloroform for three hours. The solid residue was dried and weighed (1.7 g). The filtered chloroform solution was evaporated under reduced pressure and the solid deep orange oily residue was dissolved in 1 mL of deuterated chloroform. On the three NMR tubes, thin layer chromatography was also performed on silica gel in a mixture of chloroform/methanol (9.5/0.5 *v*/*v*) solvents used as eluent. Obtained spectra together with the picture of the resulting TLC plate are reported in Figure 3.

These analyses show clearly show the purity grade of the used additives. As expected, the RGC sample is rather close in composition to the curcuminoid extract of FGC, which roughly contains 75.0% of pure curcumin (**1**) mixed with demethoxycurcumin (**2**) and bisdemethoxycurcumin (**3**).

Prior to mixing the three different additives into bitumen, thermogravimetric analyses were performed in order to check the integrity of the samples during sample preparation, but also during their eventual use in bitumen processing. TGA traces are reported in Figure 4.

As expected, the three additives are thermally stable until 200 °C, at which temperature all compounds initiate their decomposition. Note that the weight loss registered for FGC before 200 °C can be attributed to the dehydration of the turmeric powder as already observed in a previous report [25]. While both HPGC and RGC are completely calcinated, for FGC, a solid viscous white residue of ca. 8% of its weight is obtained at 850 °C. The latter is mainly formed by inorganic salts and oxides that are derived from the presence of phosphorous, potassium and magnesium in the turmeric powder (See Appendix A in ESI for SEM/EDX analysis of the solid residue).

Finally, SEM microscopy images were taken of the three samples revealing their morphological features that are reported in Figure 5. HPGC powder is characterised by nicely defined rounded parallelepipeds, pointing out the high purity and quasi-crystalline nature of the sample, while RGC presents the features of a mixture of various sized aggregates and FGC clearly shows the biological nature of the sample. EDX analysis were performed on each sample. As expected, the atomic concentration in carbon and oxygen atoms of the HQGC sample concords with the theoretical values calculated for curcumin (C: 72.43%, O: 27.57%). The EDX analysis on the FGC sample is also congruent with the biological nature of the sample, showing the characteristic elements expected for vegetal extracts. Indeed, the SEM image of FGC sample shows the expected densely packed sponge-like aggregates of various dimensions inside which the typical cellulosic walls are easily recognizable.

As regards the mechanical behaviour, a fundamental understanding of the bitumen binder’s rheological properties is essential in order to predict the end-use performance of these materials [37,38].

The visco-elastic behaviour of bitumen is characterized by its elastic (G’) and viscous (G”) moduli. These quantities, when are measured with respect to temperature, give the so-called time cure tests (temperature ramp tests). The ratio G”/G’ is tanδ. During the time cure test, as the temperature increases, bitumen starts to lose most of its G’ behaviour and begins to behave as a viscous fluid. At this point, tanδ value will exponentially increase and consequently G” will appear as the main component. Thus, the temperature at which the elastic modulus disappears and obviously tanδ → ∞ allow to unambiguously determine the sol-transition temperature t*. In cooling (inverse temperature experimental conditions), bitumen will recover its elastic feature and consequently tanδ will tend to drift towards its initial value with G’ becoming the dominant component. This therefore implies that materials with higher loss moduli have a greater ability to resist deformation while materials with higher storage moduli have a greater ability to recover from deformation [7,8,40,41,42,43].

In the following, the time cure tests of both pristine LC50 and bitumen modified with HPGC, RGC and FGC are analyzed to compare the rheological response (feedback) of the tested systems. For comparison, the effect of CaCO_3_ which is used as an inert filler (1 wt%) was also determined [44]. As expected, in line with the general trend, an increase in temperature decreases the storage modulus and increases the loss tangent.

To optimize the dosage of the additives, the LC50 bitumen was modified at 1% and 3%.

Taking into account the results shown in Table 2 and the possible impact of additives on the production costs of road pavements, 1% was chosen as the ideal dosage. In fact, at higher percentages, either no improvements are obtained, as in the case of the samples of bitumen modified with FGC, or in any case the improvements are not so significant as to justify an increase in the percentage of additive. The effect of an inert filler CaCO_3_ (1 wt%) was also determined to evaluate the effectiveness of the additive on the bitumen. Obviously, bitumen modified by the filler was mixed for 1 h under the same condition of the modified bitumen samples to evaluate also the possible oxidation phenomenon effect. Both conditions (filler and oxidation effect) did not significantly change the property of the binder and so we can say that the additive interacted with the bitumen to modify it.

In Figure 6, we report the results of all modified bitumen samples. A pronounced hardening effect was observed upon addition of the additives already dosed at 1 wt%, this was clearly evidenced by the shift of the sol-transition temperature t* observed at higher temperature values compared to that of the reference sample.

In Figure 6, the right axis is the loss tangent (tanδ). As previously described, the asymptotic value of tanδ identifies the sol-transition temperature t*. As can be observed, the investigated bio-additives show a real effect to shift the sol transition and RGC seems to have a lighter better effect compared to the other one.

A similar behavior, though with slight differences, was also recorded for C170 bitumen-based samples. In this case, while heating, t* was reached at higher values when C170 bitumen has been modified with HPGC.

It is worthy to note that the bio-additive works also on the softer bitumen.

The action mechanism of the additive and the explanation of the observed experimental differences could be due to the different chemical composition. Even though the differences are so small that this analysis, generally performed by NMR spectroscopy, is not useful at this point of this research work.

Rheological analysis, concerning C170 bitumen, gave the following results in Figure 7 and Table 3.

In Table 3, it is possible to observe closely the difference in transition temperature between the pristine C170 and modified bitumens. Moreover, in this case, the addition of a filler and the oxidation phenomenon which are due to the standard conditions of the modification did not make significant changes to the properties of the bitumen. In this case, there is a more evident change compared to the LC50 bitumen. In fact, from the results it can be seen that the difference between the bitumen + filler sample is much greater than that obtained in the analysis made on the previous bitumen (LC50). For the sample with 1% FGC, the result obtained with a longer modification time (2 h) was also reported as shown in Figure 8.

In Figure 8 and from the values of transition shown in Table 3, it is observed how the performance of the additive is also influenced by the modification time. In fact, by doubling the stirring time of the sample, an increase in performance is observed. However, this increase does not seem to be enough to justify such long times in the production phase of the modified bitumen.

We can conclude by stating that in both cases, a physico-chemical modification of the bitumen is observed. This phenomenon is more evident in the case of C170 bitumen. These differences most certainly are due to the different chemical compositions of the two types of used bitumen. In fact, they come from two crude oils and two different oil refineries.

The antioxidant properties of the additives were also tested by aging using Rolling Thin-Film Oven Test (RTFOT) according to ASTM D2872-04 [7,15]. In particular, the performance of the bitumen modified with the various additives was calculated and the transition temperature of the various samples before and after aging was calculated by means of rheological measurements (time cure test 25–120 °C). Unfortunately, none of the additives tested showed efficacy as an antioxidant agent.

### 3.1. Atomic Force Microscopy (AFM) Observation

Samples were investigated using AFM in tapping mode [45,46,47,48,49,50]. Images were acquired using probes with 150 kHz resonance frequency in air and at room temperature. Both height (left image) and phase (right image) signals were recorded simultaneously. The phase signal image, related to the mechanical properties of the specimen surface, allows to better distinguish hard domains (bright) from the elastic matrix (dark). Figure 9 refers to the pristine C170 sample. As is clearly visible, the dimensions of the domains do not exceed 4 microns in length, they are homogeneously dispersed and very few of them are interconnected. The vast majority of domains exhibit the typical bee-structure at their centre.

Sample C170 + 1% RGC shows similar features compared to the ones observed on pristine samples. The addition of RGC seems not to alter domain dimensions, as well as their arrangement on the sample surface as shown in Figure 9. The addition of HPGC has the effect of increasing the domain dimensions, that are now 5–6 μm in length even if very small hard domains less than 1 μm, are still present. On C170 + 1% HPGC sample surface, we also observe that few domains are indeed interconnected. The same effect was observed when a commercial material is used. The image shows the presence of large domains on the surface of C170 + 1% FGC sample and, as in the previous case, some of them are interconnected. This phenomenon can be attributed to the molecular structure of the additives. The dual polar/non-polar nature allows the curcumin molecule to interact with the asphaltenes with a resin-like function, already present in the bitumen. This role is fundamental in the asphaltene/maltene balance and induces the aggregations observed by the AFM experiments. All domains exhibit a well-developed bee-structure. A peculiar phenomenon is observed when the same blend undergoes 2 h of mixing instead of just 1 h, as in sample C170 + 1% FGC mixed 2 h at height temperature. In this case, domains are very small, less than 2 μm in length and in some cases, they are agglomerated in a giant domain (Figure 10).

In any case, domains, even if small in length, preserve the bee-structure and are reported in the ESI (Appendix A). AFM analysis was performed on LC50 bitumen as well, but a very peculiar and unusual phenomenon was observed. In fact, the important phase contrast cannot be observed. This structure observed suggests more in-depth studies to be carried out in other research studies to correctly describe its nature. However, the Appendix A of the ESI show some of the samples mentioned above.

### 3.2. Adhesion Test

Table 4 and Appendix A in ESI show the results of the boiling tests [13,51,52] of the samples at different percentages of additive with a stone (porphyry) where the bitumen LC50 does not show good adhesion [53,54]. The chemical analysis of this type of stone is reported in the following article [7].

Moreover, in this case, it is observed that percentages higher than 1% of additive do not confer significant improvements. HPGC and RGC additives work very well as adhesion activators, while FGC improves the adhesion between bitumen and stone but much less than the other additives analysed (only 35–40%). Taking into consideration the best result obtained was with 3% (which is a percentage three times higher than that used for the other two additives), it does not achieve results comparable to the latter.

This phenomenon could be due to chemical composition of the FGC additive. FGC is a commercial turmeric. The high percentage of cellulose, present in the mixture, limits the adhesion properties of the additive but according to literature data [36], significantly increases the mechanical behaviour of the binder.

Table 5 and Appendix A in ESI show the results of the boiling tests of the tested samples using also in this case, a stone (porphyry) with which the C170 bitumen already shows a good affinity. Given the results obtained from LC50, only the modified bitumen samples with 1% of the various additives were tested. As already mentioned, since bitumen already has an affinity to this type of stone, we cannot hold in high regard the significant differences between the various samples. We can only say that in this case, only HPGC and FGC improved the adhesion between the bitumen and the stones.

## 4. Conclusions

In this report, we investigated the effects that turmeric spice, used as an additive, might induce bitumen’s chemico-physical properties. In particular, we investigated its efficiency as a rheological modifier, adhesion promoter and antioxidant agent. Furthermore, we investigated the effect of curcumin, the main bioactive compound of turmeric, compound known for its antioxidant properties. The results showed that turmeric and curcumin are very efficient as adhesion promoters and as rheological modifiers while none of them works as an antioxidant agent. The different performances and efficiencies obtained for each additive with the two types of bitumen used highlight that their effectiveness is strongly influenced by the chemical nature of the neat bitumen. The surfactant nature of the additives plays a role similar to the resins normally present in bitumen. These molecules act on the molecular structure of the asphaltene modifying the supramolecular organization. This effect is reflected in the improved mechanical performances of the modified bitumens. The use of food grade additives will facilitate a reduction in several forms of pollution from affiliated industrial processes in a bid to help protect the environment. The study presented herein shows the possibility of using such economically advantageous food grade additives incorporated in small amounts to improve the rheological and adhesion properties of bitumen. Further work is also in progress to identify the right natural antioxidant.

## Figures and Tables

**Figure 1 materials-14-01622-f001:**
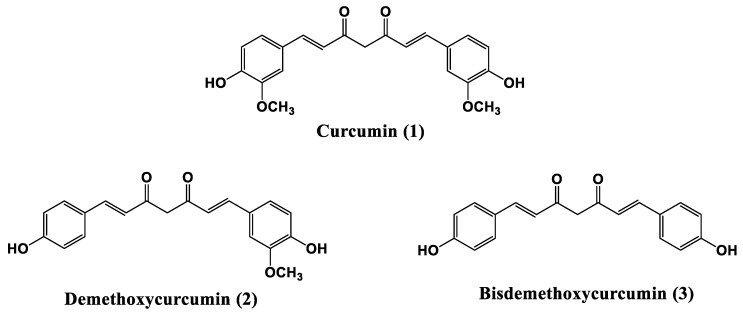
Structures of curcuminoids.

**Figure 2 materials-14-01622-f002:**
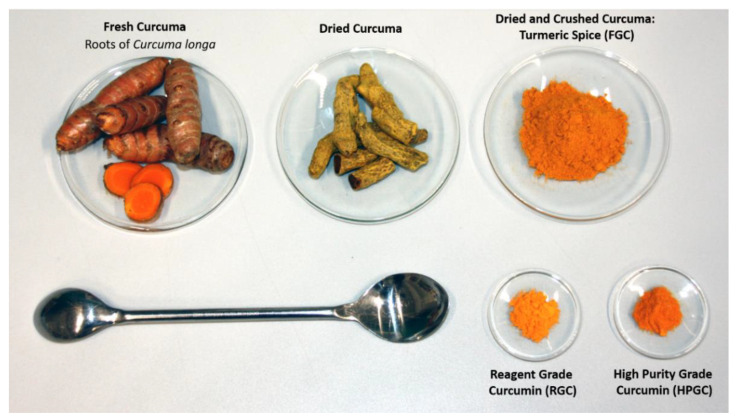
Fresh and dry curcuma rhizomes and resulting turmeric powder (FGC), together with the two differently pure grades of curcumin used in this work (RGC and HPGC).

**Figure 3 materials-14-01622-f003:**
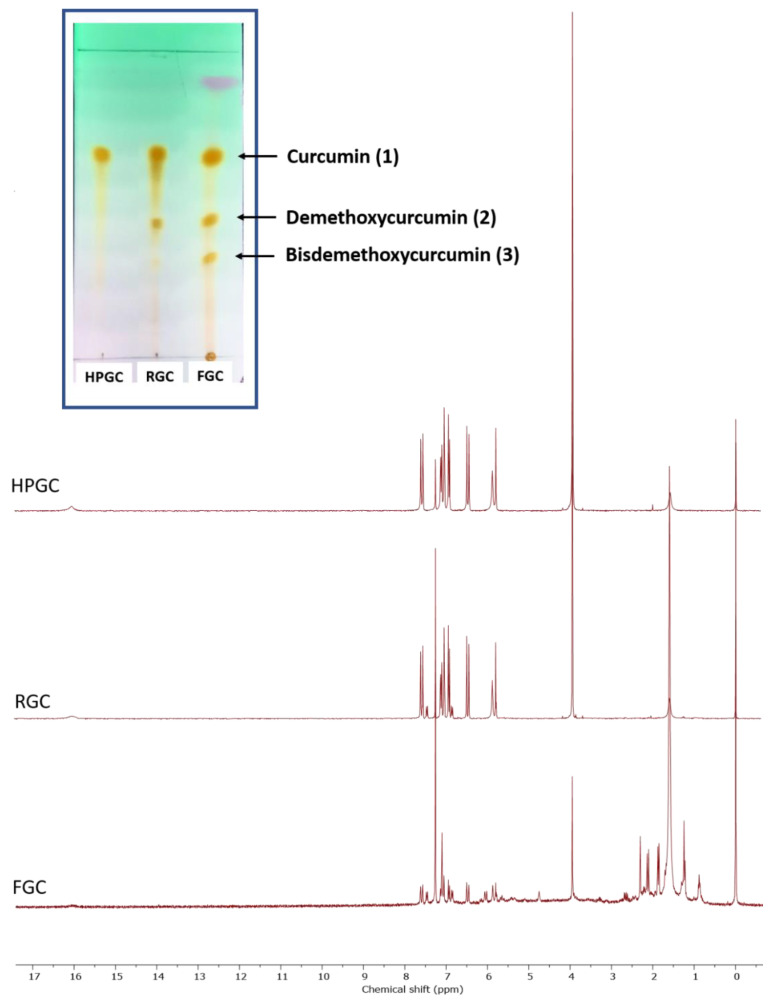
^1^H-NMR spectra in CDCl_3_ of HPGC, RGC and chloroform extracted FGC. In inset photograph of a TLC plate under UV-light (eluent: CHCl_3_/MeOH 9.5/0.5 *v*/*v*).

**Figure 4 materials-14-01622-f004:**
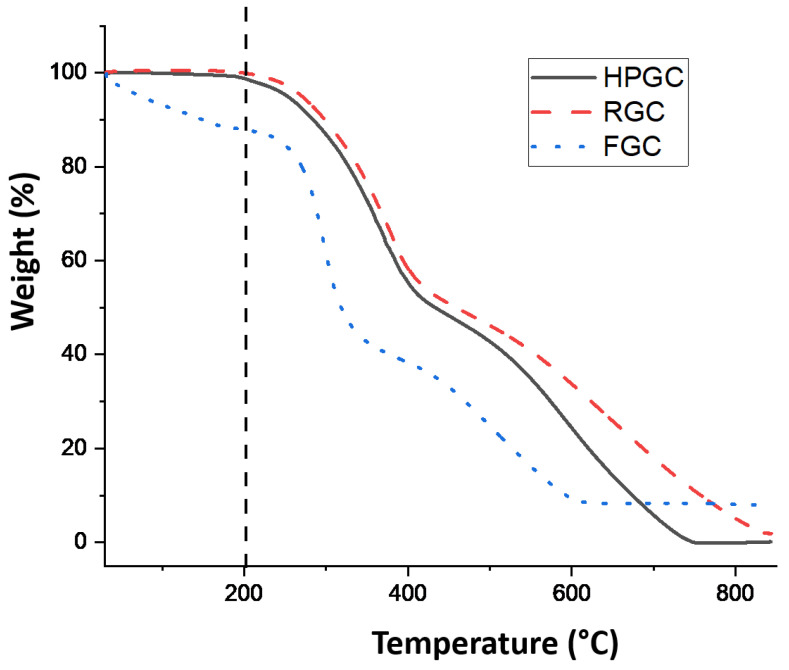
TGA traces of HPGC, RGC and FGC samples.

**Figure 5 materials-14-01622-f005:**
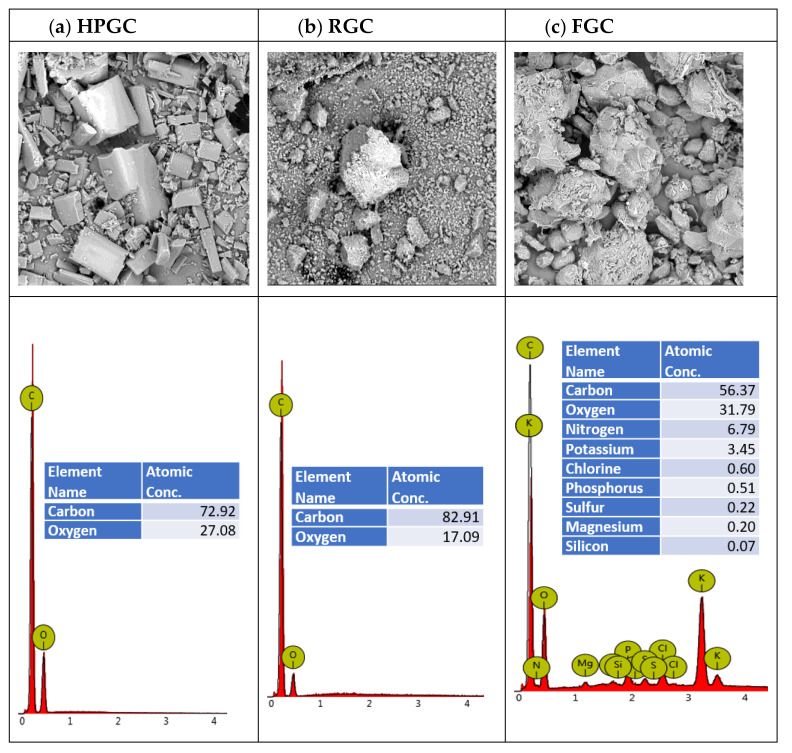
SEM images and relative composition determined by in situ EDX analysis of (**a**) HPGC, (**b**) RGC and (**c**) FGC. Picture dimensions: 500 μm × 500 μm.

**Figure 6 materials-14-01622-f006:**
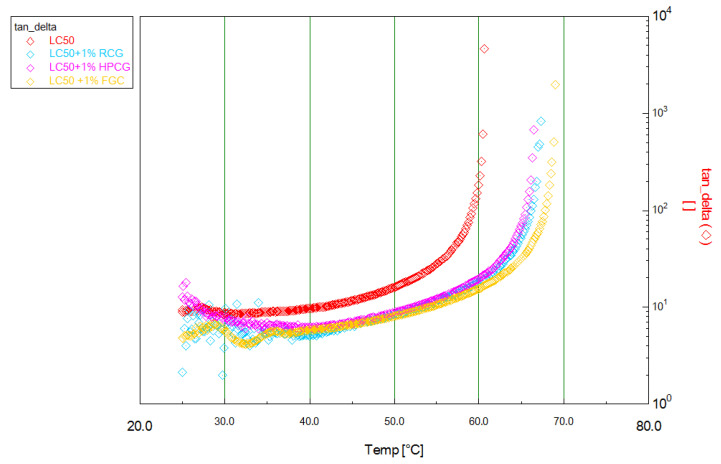
Time Cure Test 1—Semi-log plot of temperature ramp tests for the pristine LC50 and modified bitumen formulated with 1 wt% of bioadditive added as solid powder dispersion.

**Figure 7 materials-14-01622-f007:**
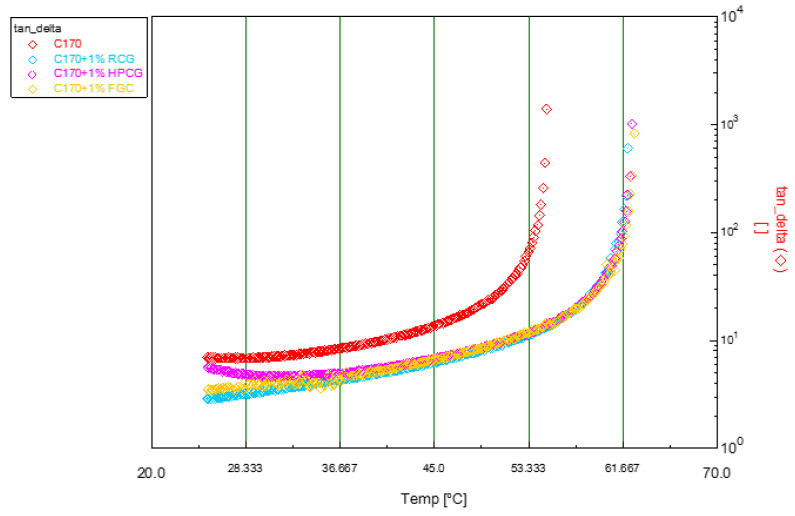
Time Cure Test—Semi-log plot of temperature ramp tests for the pristine C170 and modified bitumen formulated with 1 wt% of bioadditive as solid powder dispersion.

**Figure 8 materials-14-01622-f008:**
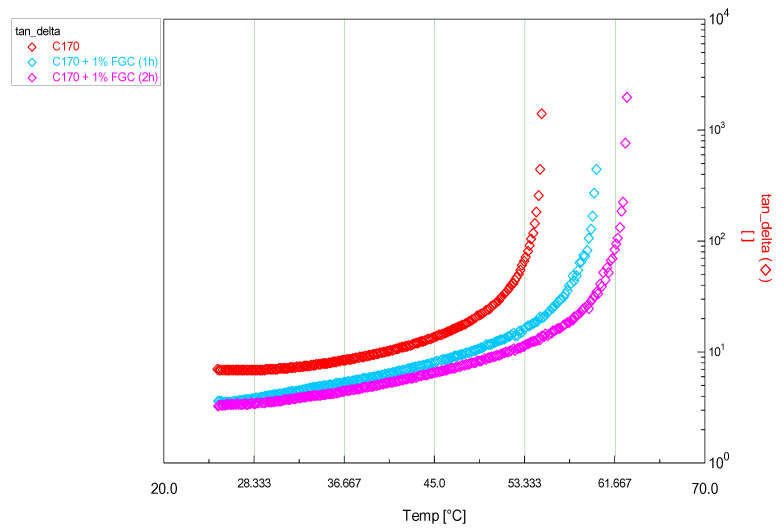
Time Cure Test—Differences in transition temperature between the same sample undergoing an extra hour of mixing at 150 °C.

**Figure 9 materials-14-01622-f009:**
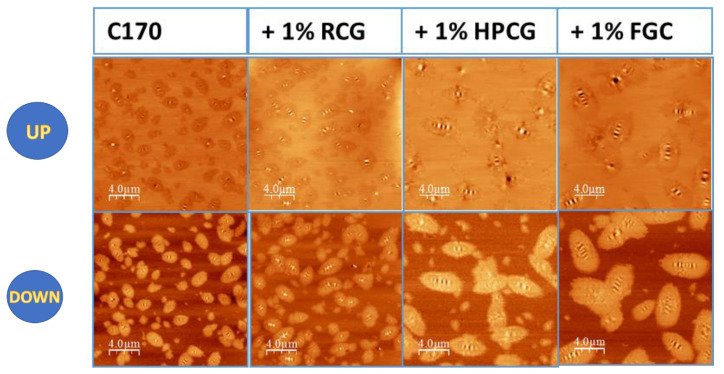
Sample surface: topography (**UP**), phase image (**DOWN**) Bitumen C170 Pure and Modified.

**Figure 10 materials-14-01622-f010:**
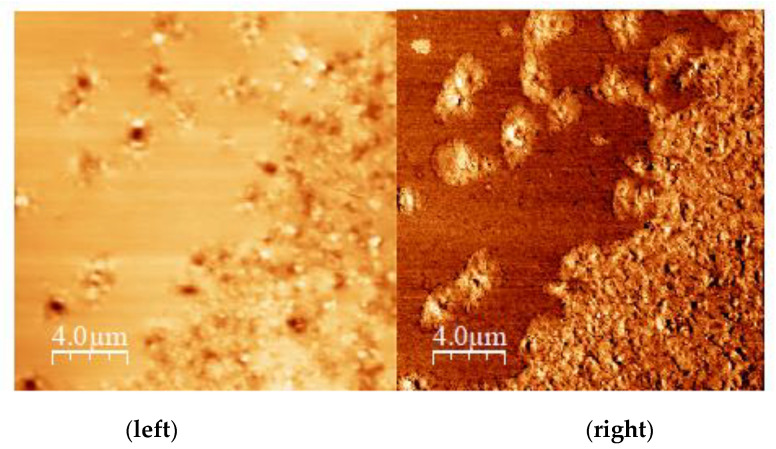
FGC 2 h sample surface: topography (**left**), phase image (**right**).

**Table 1 materials-14-01622-t001:** Fundamental physical properties of bitumens.

Measured Properties	Standard	Unit	LC50	C170
			50/70	170/210
Penetration at 25 °C	EN 1426	0.1 mm	68 ± 1	185 ± 1
Softening point (R&B)	EN 1427	°C	48.8 ± 0.2	42.6 ± 0.2
Flash point	EN 2592	°C	≥230	≥220
Solubility	EN 12592	% (m/m)	≥99	≥99

**Table 2 materials-14-01622-t002:** Difference in transition temperature.

Sample	Transition Temperature (°C) ±0.1	∆ (°C) ±0.2	∆ (%)
LC50	60.5	-	-
LC50 + 1% CaCO_3_	65.0	4.5	7.4
LC50 + 1% FGC	66.6	6.1	10.1
LC50 + 3% FGC	66.5	6.0	9.9
LC50 + 1% RGC	67.5	7.0	11.6
LC50 + 3% RGC	70.1	9.6	15.9
LC50 + 1% HPGC	65.5	5.0	8.3
LC50 + 3% HPGC	67.5	7.0	11.6

**Table 3 materials-14-01622-t003:** Difference in transition temperature between the pure sample and the various modified compounds.

Sample	Transition Temperature (°C) ±0.1	Δ (°C) ±0.2	Δ (%)
C170	51.1	-	-
1% HPGC	62.2	11.1	21.7
1% RGC	60.9	9.8	19.2
1% FGC (1h)	61.5	10.4	20.4
1% FGC (2h)	62.3	11.2	21.9
1% CaCO_3_	55.0	3.9	7.6

**Table 4 materials-14-01622-t004:** Boiling test results for LC50.

Sample	Coverage (%)
PURE LC50	15
LC50 + 1% RGC	80
LC50 + 3% RGC	80
LC50 + 1% HPGC	75
LC50 + 3% HPGC	75
LC50 + 1% FGC	35
LC50 + 3% FGC	40

**Table 5 materials-14-01622-t005:** Boiling test results for C170.

Sample	Coverage (%)
PURE C170	70
C170 + 1% RGC	75
C170 + 1% HPGC	90
C170 + 1% FGC	90

## Data Availability

The data presented in this study are available on request from the corresponding authors.

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
