# Peer review of "Spicy Bitumen: Curcumin Effects on the Rheological and Adhesion Properties of Asphalt"

_materials, 2021, doi:10.3390/ma14071622_

Round 1
Reviewer 1 Report
The paper deserves publication. Nevertheless, some minor changes could aid to improve it:
- In the introduction section, the authors state that the current common additives are not eco-friendly. It is true, but some other additives exist that are eco-friendly, such as, for example some biopolymers (lignin, suberin…). Please, update the state of the art.
- In figure 4, please, use other styles to better differentiate HPGC, RGC and FGC.
- In page 8, line 236 clarify to the readers this sentence: “FCG clearly shows the biological nature of the sample”. Maybe for the authors is clear, but for some readers could be not so obvious.
- The authors used 1% and 3% of additives by weight of bitumen. Why by weight and not by volume?
- In page 10, lines 278 to 291. Please, rewrite. 1% FGC and 3% FCG display similar results, but 1% RGC or HPGC and 3% of the same material, are no so similar.
- In the same page and in the same lines, the authors state that the effect of the filler and the effect of the oxidation did not significantly change the properties of the binder. But in table 2, how were obtained the results for the LC50? Were they obtained on a bitumen without heating it or not? That is, please, clarify if the LC50 has been submitted to the same experimental conditions than the rest of the cases. If this bitumen (LC50) has been submitted to these same conditions, how can the authors state that the effect of the oxidation is not significant? Tests on LC50 without conditioning are needed. On the contrary, if the LC50 has not been submitted to these same conditions, explain in more detail why the authors can state that the effect of the filler and the oxidation is not significant, because a 7.4% of change occurs (is this percentage not significant?) Please, rewrite this paragraph taking into account all these comments.
- In page 13 line 350 is correct left image and right image? Or it must be changed by up and down?
- Is it possible to include a new figure S7 with higher quality?
Author Response
Referee 1
Comments and Suggestions for Authors
The paper deserves publication. Nevertheless, some minor changes could aid to improve it:
- In the introduction section, the authors state that the current common additives are not eco-friendly. It is true, but some other additives exist that are eco-friendly, such as, for example some biopolymers (lignin, suberin…). Please, update the state of the art.
We thank the referee for the suggestions. We changed the introduction updating the literature.
Here is the revised text:
The aforementioned additives, even though proven to be effective, are generally not eco-friendly as they are either too acidic, too basic or in one way or another, adversely affect the environment in which the road is paved. Though recently some bio-polymers are becoming tested as modifiers in road paving [ref]
- Pérez Pérez,Ana Marí, Rodríguez Pasandín, Jorge Carvalho Pais, Paulo Antonio Alves Pereira, Use of lignin biopolymer from industrial waste as bitumen extender for asphalt mixtures. Journal of Cleaner Production Volume 220, 20 May 2019, Pages 87-98.
- Mohsen Zahedi, Ali Zarei, Mohammad Zarei. The effect of lignin on mechanical and dynamical properties of asphalt mixtures SN Applied Sciences (2020) 2:1242 | https://doi.org/10.1007/s42452-020-3041-4
- In figure 4, please, use other styles to better differentiate HPGC, RGC and FGC.
Fig. 4 has been replaced. Different styles have been used to better differentiate the three components of this graph, increasing furthermore the width of the traces for easy reading.
- In page 8, line 236 clarify to the readers this sentence: “FCG clearly shows the biological nature of the sample”. Maybe for the authors is clear, but for some readers could be not so obvious.
A sentence has been added to the main text to describe the vegetal features of the FGC SEM image.
“indeed the SEM image of FGC sample shows densely packed sponge-like aggregates of various dimensions inside which the typical cellulosic walls are easily recognized”.
- The authors used 1% and 3% of additives by weight of bitumen. Why by weight and not by volume?
In the bitumen world, it’s common to use %wt. We think this exists for practical reasons.
Generally, the additives in the road pavement sector and in works of scientific literature are added in %weight of bitumen:
- Link: https://www.sasobit.com/files/downloads/en/sasobit/DosageRecommendations_en.pdf
- Ashok Julaganti, Rajan Choudhary & Abhinay Kumar, Rheology of modified binders under varying doses of WMA additive–Sasobit Petroleum Science and Technology Volume 35, 2017 - Issue 10.
- Alim Feizrakhmanovich Kemalov, Ruslan Alimovich Kemalov, Ilmira Maratovna Abdrafikova, Pavel Sagitovich Fakhretdinov, and Dinar Zinnurovich Valiev, Polyfunctional Modifiers for Bitumen and Bituminous Materials with High Performance. Advanced Pavement Materials for Sustainable Transportation Infrastructure Volume 2018 |Article ID 7913527 | https://doi.org/10.1155/2018/7913527.
- Special Tender Specification. In Road Paving; Technical Standard 2009.
- In page 10, lines 278 to 291. Please, rewrite. 1% FGC and 3% FCG display similar results, but 1% RGC or HPGC and 3% of the same material, are no so similar.
Tripling the dosage of RCC and HPGC would have not been convenient (both in economic and performance terms) considering the fact that although they are not the same, the differences are not so pronounced. Practically, the lowest possible dosage with the best performance is always used.
- In the same page and in the same lines, the authors state that the effect of the filler and the effect of the oxidation did not significantly change the properties of the binder. But in table 2, how were obtained the results for the LC50? Were they obtained on a bitumen without heating it or not? That is, please, clarify if the LC50 has been submitted to the same experimental conditions than the rest of the cases. If this bitumen (LC50) has been submitted to these same conditions, how can the authors state that the effect of the oxidation is not significant? Tests on LC50 without conditioning are needed. On the contrary, if the LC50 has not been submitted to these same conditions, explain in more detail why the authors can state that the effect of the filler and the oxidation is not significant, because a 7.4% of change occurs (is this percentage not significant?) Please, rewrite this paragraph taking into account all these comments.
We thank the referee. We agree that our discussion was not so clear. The LC50 sample is a pristine bitumen analyzed without any additive or oxidation process. The bitumen modified with filler allows us to evaluate the real effectiveness of the additives. In fact, using this, we can estimate oxidation and filler effects simultaneously.
We modified the text in the revised version
- In page 13 line 350 is correct left image and right image? Or it must be changed by up and down?
We thank the referee. The figures are correct. In figure 9, the top side represents the topography of all investigated samples while in the figure 10 the topography is represented by the left image and it is the AFM analysis of only one sample.
- Is it possible to include a new figure S7 with higher quality?
We did this according the referee’s request.

Reviewer 2 Report
- Very nice title of your paper.
- In the abstract you have stated you have used DSR, but I cant find it in the manuscript.
- I have doubt about your insistence on writing about sustainable and ecological side of your modifiers. As far as I know curcuma requires high humidity, which results in high water consumption. Especially if it would be required in the high amount to modify asphalt.
- Please improve the conclusions part. It is kind of weak.
Author Response
Referee 2
- Very nice title of your paper.
- In the abstract you have stated you have used DSR, but I cant find it in the manuscript.
We evidenced the DSR better in the experimental section.
- I have doubt about your insistence on writing about sustainable and ecological side of your modifiers. As far as I know curcuma requires high humidity, which results in high water consumption. Especially if it would be required in the high amount to modify asphalt.
The additives commonly used in the road pavement sector are very toxic. They are generally strongly acidic, strongly basic, aromatic compounds etc so the possibility to use a food-grade additive as a matter of fact is a nice ecological innovation in the road pavement sector.
We do not really understand the point made by the referee. Maybe the referee is thinking of the amount of water required to grow the plant in large quantity. In this case, we do think that using a green modifier obtained from cultivation is still better for the environment than chemicals dispersed into bitumen. These chemicals are mostly obtained from industries and they connote a high risk of pollution to the environment and fauna both during manufacturing processes and the deposition on the ground during road pavement processes.
- Please improve the conclusions part. It is kind of weak.
We improved the conclusion according to suggestion of the referee.

Reviewer 3 Report
Review
Title: “Spicy Bitumen: Turmeric and Curcumin effects on the rheological and adhesion properties of Asphalt”
This is a novel article focusing on the effects of curcumin on Asphalt properties. The manuscript describes the effectiveness of a food grade bio-additive on three properties, namely: (a) physico-chemical properties such as transition temperature of asphalt binder, (b) the affinity between binder and stone aggregates, and (c) the effects of oxidation caused by exposure to air, water and other natural elements during the production of asphalt pavements. The mechanisms by which the additive confers these desired features on bitumen is hypothesized. The study was performed through Dynamic Shear Rheology (DSR), Atomic Force Microscopy, Scanning Electron Microscopy (SEM) and Boiling Test Analysis.
In general, the manuscript is well written and presents interesting findings. I am not convinced about whether the addition of turmeric even in small portions will be economically viable, but nevertheless it is an interesting study, that I consider it should be published.
It is strongly advised to edit the language of the
Following are some editing suggestions:
Title: I would suggest erasing the term “turmeric”. Essentially you are describing the same material, since turmeric is the commercial name of curcumin.
The title suggests that they are two different materials.
Line 111. “Owing to the multi-components nature of Turmeric (Food Grade Curcumin, FGC), in order to check the eventual role of curcuminoids onto the properties modifications of bitumen, we also investigated the modifications brought by using two differently pure grades of curcumin, allowing us to determine the functionality and effectiveness of each one.”
You could change it to:
“Owing to the multi-component nature of Turmeric (Food Grade Curcumin, FGC) and in order to examine the effects of curcuminoids on the bitumen properties, two different pure grades of curcumin were also investigated. Thus, the functionality and effectiveness of each one was determined.”
Line 134 change “bitumen” to “bitumens”
Line 142 change to “CaCO3”
Line 225 change to: “Note that the weight loss registered for FGC before 200°C can be attributed to the dehydration of the turmeric powder as already observed in a previous report”
Line 247 change to “an understanding of the rheological properties of binder” to “an understanding of the binder’s rheological properties”
Figures 6 and 7. There are no arrows shown in the Figures. Moreover, the legends need to be short. More the description of the Figures in the text.
Line 320 you mention: “the pure sample and the various modified compounds” but elsewhere in the text you use the term pristine C170 and modified bitumens. You should use the same terminology everywhere in the text.
Line 361 “The adding of RGC seems not alter domain dimensions as well as their arrangement on the sample surface as shown in Figure 9” change to “The addition of RGC seems not to alter domain dimensions, as well as their arrangement on the sample surface as shown in Figure 9.”
In the section 3. Results and Discussion. The authors in many cases merely mention the results, without providing possible explanations for the findings.
Line 368: “A peculiar phenomenon is observed when the same blend undergoes 2 hours of mixing instead of just 1 hour, as in sample C170+1% FGC mixed 2h at height temperature.”
Why do you believe this happens? The longer mixing duration affects the FGC?
Line 389. Please rewrite, it is not clear: “Taking into consideration the best result obtained was with 3% (which is a percentage 3 times higher than that used for the other 2 additives), it does not achieve results comparable to the latter.”
Line 410 “The results showed that turmeric and curcumin are” why turmeric and curcumin are identified as two different modifiers?
Line 416 “The study presented herein shows the possibility of using such food grade additives economically advantageous and in small amount to improve the rheological and adhesion properties of bitumen and further work are in progress to identify the right natural antioxidant.” Syntactic error, please rewrite. Try to keep the sentences as small as possible.
Moreover, you use the term “economically advantageous” but you do not mention in the text anything about costs. You should not draw conclusions without prior discussion.
Author Response
Referee 3
This is a novel article focusing on the effects of curcumin on Asphalt properties. The manuscript describes the effectiveness of a food grade bio-additive on three properties, namely: (a) physico-chemical properties such as transition temperature of asphalt binder, (b) the affinity between binder and stone aggregates, and (c) the effects of oxidation caused by exposure to air, water and other natural elements during the production of asphalt pavements. The mechanisms by which the additive confers these desired features on bitumen is hypothesized. The study was performed through Dynamic Shear Rheology (DSR), Atomic Force Microscopy, Scanning Electron Microscopy (SEM) and Boiling Test Analysis.
In general, the manuscript is well written and presents interesting findings. I am not convinced about whether the addition of turmeric even in small portions will be economically viable, but nevertheless it is an interesting study, that I consider it should be published.
- It is strongly advised to edit the language of the
The manuscript was revised by a native English speaker.
- Following are some editing suggestions:
- Title: I would suggest erasing the term “turmeric”. Essentially you are describing the same material, since turmeric is the commercial name of curcumin.
- The title suggests that they are two different materials.
We understand the referee but in our opinion they clearly are two different materials as explained in the main text. Maybe the referee could be confused about the terms curcuma and curcumin. Curcuma is the name of the plant, sometimes used as the name of the spice (in some countries) but most of the time, the spice is referred to as Turmeric powder, and for this reason Turmeric/curcuma is a mixture of several components - the major part of it being of cellulose nature as declared in the main text. Curcumin instead is a molecule of define molecular weight that is contained in curcuma/turmeric and is the major component of the curcuminoids present in the rhizome of the plant. For this reason, we decided to maintain in the title the two names since they clearly are different materials.
- Line 111. “Owing to the multi-components nature of Turmeric (Food Grade Curcumin, FGC), in order to check the eventual role of curcuminoids onto the properties modifications of bitumen, we also investigated the modifications brought by using two differently pure grades of curcumin, allowing us to determine the functionality and effectiveness of each one.”
- You could change it to:
“Owing to the multi-component nature of Turmeric (Food Grade Curcumin, FGC) and in order to examine the effects of curcuminoids on the bitumen properties, two different pure grades of curcumin were also investigated. Thus, the functionality and effectiveness of each one was determined.”
We thank the referee for this suggestion which has been accepted. The Main text has therefore been modified.
- Line 134 change “bitumen” to “bitumens”
We did this according to the referee’s request
- Line 142 change to “CaCO3”
We did this according to the referee’s request
- Line 225 change to: “Note that the weight loss registered for FGC before 200°C can be attributed to the dehydration of the turmeric powder as already observed in a previous report”
We thank the referee for this suggestion which has been accepted. The Main text has therefore been modified.
- Line 247 change to “an understanding of the rheological properties of binder” to “an understanding of the binder’s rheological properties”
We did this according to the referee’s request.
- Figures 6 and 7. There are no arrows shown in the Figures. Moreover, the legends need to be short. More the description of the Figures in the text.
We changed the figures according to the suggestions of the referee.
- Line 320 you mention: “the pure sample and the various modified compounds” but elsewhere in the text you use the term pristine C170 and modified bitumens. You should use the same terminology everywhere in the text.
We changed the text according to the suggestions of the referee.
- Line 361 “The adding of RGC seems not alter domain dimensions as well as their arrangement on the sample surface as shown in Figure 9” change to “The addition of RGC seems not to alter domain dimensions, as well as their arrangement on the sample surface as shown in Figure 9.”
We did this according to the request of the referee
- In the section 3. Results and Discussion. The authors in many cases merely mention the results, without providing possible explanations for the findings.
We added more explanations in the text.
- Line 368: “A peculiar phenomenon is observed when the same blend undergoes 2 hours of mixing instead of just 1 hour, as in sample C170+1% FGC mixed 2h at height temperature.”
- Why do you believe this happens? The longer mixing duration affects the FGC?
We agree with the referee. We modified the text trying to explain the reasons behind this phenomenon
- Line 389. Please rewrite, it is not clear: “Taking into consideration the best result obtained was with 3% (which is a percentage 3 times higher than that used for the other 2 additives), it does not achieve results comparable to the latter.”
We did this according to the referee’s suggestion
- Line 410 “The results showed that turmeric and curcumin are” why turmeric and curcumin are identified as two different modifiers?
We understand the referee but in our opinion they clearly are two different materials as explained in the main text. The referee could be confused between the terms curcuma and curcumin. Curcuma is the name of the plant, sometimes used as the name of the spice (in some countries) but most of the time, the spice is referred as Turmeric powder, and for this reason Turmeric/curcuma is a mixture of several components the major part of it being of cellulose nature as declared in the main text. Curcumin instead is a molecule of define molecular weight that is contained in curcuma/turmeric and is the major component of the curcuminoids present in the rhizome of the plant.
- Line 416 “The study presented herein shows the possibility of using such food grade additives economically advantageous and in small amount to improve the rheological and adhesion properties of bitumen and further work are in progress to identify the right natural antioxidant.” Syntactic error, please rewrite. Try to keep the sentences as small as possible.
- Moreover, you use the term “economically advantageous” but you do not mention in the text anything about costs. You should not draw conclusions without prior discussion.
Our idea which is of ​​economic advantage refers not to the use of food grade products as in their initial original state but to the use of production waste derived from the food production processes. These wastes are therefore no longer materials to be disposed of but become resources by encouraging the circular economy of the system. We tested the pure product to analyse the action of the active ingredient in modifying the bitumen.

Round 2
Reviewer 2 Report
Thank you for addressing my comments.
Author Response
Referee 2
English language and style are fine/minor spell check required
The manuscript was revised by researcher whose mother tongue is English
Reviewer 3 Report
The authors addressed most of the comments, and as a result the manuscript has been significantly improved.
However, there are still some aspects that need to be addressed:
You mention that turmeric and curcumin are different. And I won’t argue about that. Nonetheless, you examined three different additives (HPGC, RGC and FGC) that basically all contain curcumin in different percentages. You do mention in your reply “Curcumin instead is a molecule of define molecular weight that is contained in curcuma/turmeric and is the major component of the curcuminoids present in the rhizome of the plant.”
Hence, from my point of view you actually tested the effect that curcumin has on the bitumen’s properties.
As a result, I believe that a better title would be: “Spicy Bitumen: Curcumin effects on the rheological and adhesion properties of Asphalt”
Line 142-150: What is the purity for HPLC in the examined turmeric?
Other than that I have no further suggestions.
Author Response
Referee 3
The authors addressed most of the comments, and as a result the manuscript has been significantly improved.
However, there are still some aspects that need to be addressed:
You mention that turmeric and curcumin are different. And I won’t argue about that. Nonetheless, you examined three different additives (HPGC, RGC and FGC) that basically all contain curcumin in different percentages. You do mention in your reply “Curcumin instead is a molecule of define molecular weight that is contained in curcuma/turmeric and is the major component of the curcuminoids present in the rhizome of the plant.”
Hence, from my point of view you actually tested the effect that curcumin has on the bitumen’s properties.
As a result, I believe that a better title would be: “Spicy Bitumen: Curcumin effects on the rheological and adhesion properties of Asphalt”
We are still of our idea that Turmeric and Curcumin are two different materials, but we do accept to change the title of the article as requested by the referee.
Line 142-150: What is the purity for HPLC in the examined turmeric?
Here, we reported exactly what the supplier reported in the data sheet of the product, however a “for” was added by mistake. This has been corrected.
Other than that I have no further suggestions.